# In Situ Acoustic Treatment of Anaerobic Digesters to Improve Biogas Yields

**John Loughrin \*** , **Stacy Antle, Karamat Sistani and Nanh Lovanh**

United States Department of Agriculture, Agricultural Research Service, Food Animal Environmental Systems Research Unit, 2413 Nashville Road, Suite B5, Bowling Green, KY 42101, USA; stacy.antle@usda.gov (S.A.); karamat.sistani@usda.gov (K.S.); nanh.lovanh@usda.gov (N.L.)

\* Correspondence: john.loughrin@usda.gov; Tel.: +1-270-781-2260

**Abstract:** Sound has the potential to increase biogas yields and enhance wastewater degradation in anaerobic digesters. To assess this potential, two pilot-scale digestion systems were operated, with one exposed to sound at less than 10 kHz and with one acting as a control. Sounds used were sine waves, broadband noise, and orchestral compositions. Weekly biogas production from sound-treated digesters was 18,900 L, more than twice that of the control digester. The sound-treated digesters were primarily exposed to orchestral compositions, because this made cavitational events easier to identify and because harmonic and amplitude shifts in music seem to induce more cavitation. Background recordings from the sound-treated digester were louder and had more cavitational events than those of the control digester, which we ascribe to enhanced microbial growth and the resulting accelerated sludge breakdown. Acoustic cavitation, vibrational energy imparted to wastewater and sludge, and mixing due to a release of bubbles from the sludge may all act in concert to accelerate wastewater degradation and boost biogas production.

**Keywords:** acoustic cavitation; anaerobic digestion; biogas; green energy; methane; sonication

## 1. Introduction

Anaerobic digestion, in simplified form, is the process wherein organic matter is broken down in an oxygen-free environment by the combined processes of fermentation and methanogenesis to produce biogas, a mixture of carbon dioxide and methane, in addition to trace gases such as hydrogen sulfide and ammonia [1]. As a crude form of natural gas, biogas represents both a resource and a means of limiting emissions from wastewater treatments and animal rearing operations [2].

Anaerobic digestion, however, is typically slow, limited by slow sludge hydrolysis and the slow growth of anaerobic microorganisms [3,4]. This is usually treated by operating the digesters at elevated temperatures to improve the rate of treatment and biogas production [3].

Recently, we investigated if sound could also be used to enhance anaerobic digestion by accelerating sludge breakdown and thereby improve biogas yields [5]. Phenomena that sound waves could induce in wastewater include, but are not limited to, mechanical vibration of the wastewater and sludge [6], acoustic streaming of the liquid to enhance nutrient mixing [7], and induction of acoustic cavitation in which bubbles oscillate in the presence of an applied acoustic field [8,9].

This experiment was conducted by contrasting the performances of control and sound-treated digesters in three 100-day trials [5]. The sound-treated digesters were exposed to sound at sonic frequencies (<20,000 Hz) by means of underwater speakers. During these trials, the digesters were for the most part exposed to either sine waves at 1000 Hz or up five sine waves, at 1000 Hz harmonics up to 5000 Hz.

Digesters exposed to sound produced approximately 12% more biogas and had significantly less chemical oxygen demand (COD) than did the control digesters. Furthermore, sludge from the digesters exposed to sound was more degraded than that of control digesters. Although we could not conclusively determine the mechanisms involved, we did observe bubble harmonics indicative of acoustic cavitation (as discussed below), and at infrequent intervals, broad-band noise indicative of shock waves propagated from cavitation cloud collapse [5].

Other forms of cavitation have been used to enhance wastewater breakdown and biogas production. Hydrodynamic cavitation, also known as inertial cavitation, has been exploited to remediate wastewater [10,11]. In hydrodynamic cavitation, cavitation inception happens in regions of low pressure such as occur at the outlets of restricted flows or from propellers [10]. The relatively low-pressure bubbles that form then collapse as they migrate to regions of higher pressure. In many situations, the damage induced by this cavitation can cause significant wear to mechanical parts and has long been the subject of intense research.

In a related manner, ultrasonification (>20 kHz) of waste has also been used as a pretreatment to help disintegrate sludge particles, thereby enhancing wastewater degradation and biogas production [6,12,13]. This is a form of noninertial cavitation, wherein a bubble oscillates in the presence of an acoustic field. As a pretreatment, this differed from our research, wherein sound at sonic frequencies (<20 kHz) was employed and treatment occurred in the digestion tank.

As bubbles in an acoustic field oscillate at frequencies proportional to their radii, gases from the surrounding liquid will tend to flow into bubbles as their radii increase and their internal pressures drop; or, as radii decrease and internal pressures increase, there will be a tendency for gas efflux from the bubbles to occur [14,15]. Although acoustic cavitation is often referred to as "stable" cavitation since it involves a repetitive cycle of bubble oscillations [8], bubbles oscillating in the presence of an acoustic field will tend to grow to a point where collapse occurs as their internal pressures drop to a value below that of the vapor pressure in the surrounding liquid. Acoustic cavitation typically causes less localized damage than hydrodynamic cavitation does. Furthermore, sound at sonic frequencies may travel great distances, depending on frequency and contingent on scattering and attenuation due to suspended solids and bubbles [16,17].

During acoustic cavitation, a bubble excited at a frequency $f_0$ will resonate at harmonics $2f_0$, $3f_0, 4f_0, \ldots, nf_0$, with n representing an indeterminate positive integer, and at ultraharmonic frequencies $^3/_2f_0, \; ^5/_2f_0, \; ^7/_2f_0, \ldots \; ^m/_2f_0$, in which m represents an indeterminate odd-numbered positive integer. The first series represents stable or acoustic cavitation with linear bubble oscillations, whereas the latter series is indicative of chaotic oscillations and incipient bubble collapse. In the earlier investigation [5], the appearance of ultraharmonic peaks was a prelude to broadband noise, indicating widespread cavitation cloud collapse [9]. The acoustic emissions due to bubble harmonics, ultraharmonics, and cavitational collapse may be quite intense. For these reasons, bubbles could be considered a source of potential energy, in which the benefits of acoustically stimulating a highly gaseous liquid are perhaps disproportionate to the energy input required.

Biogas production decreased with higher waste loading and the resulting higher suspended solids. We hypothesized that this might be due to scattering and absorption of sound within the sludge layer, limiting the effectiveness of the treatment. Therefore, sonic treatment might work more effectively at larger scales, since this would allow for a more favorable placement of the speakers above the sludge layer and less interference with sound transmission throughout the digester. Furthermore, as pressure increases with depth, the internal pressure required to maintain bubble integrity at a given radius will necessarily increase. Therefore, the forces generated by cavitational collapse will be greater. Both factors could increase the effectiveness of sonic treatments and enhance biogas yields. Here, a system employing sonic technology in an anaerobic digester at pilot-scale is described and compared to a control system.

## 2. Materials and Methods

### 2.1. Waste Treatment System Design

The system was constructed on property leased by the Agricultural Research Service and located on the farm of Western Kentucky University. In describing the construction of the anaerobic digestion system, English (Imperial) systems of measurement are used to describe supplies and equipment, as per manufacturer's specifications.

The experimental system (Figure 1) consisted of two parallel rows of high-density polyethylene (HDPE) water tanks (Plastic Mart, Austin, TX, USA,), each consisting of a 3000 gallon (11,360 L) primary digester with a diameter of 2.44 m by 2.77 m tall; 1000 gallon (3785 L) secondary anaerobic digester (1.63 m diameter by 2.03 m tall); 800 gallon (3028 L) partial aeration tank (1.63 m diameter by 1.85 m tall); and 305 gallon (1150 L) holding tank (1.17 m diameter by 1.27 m tall).

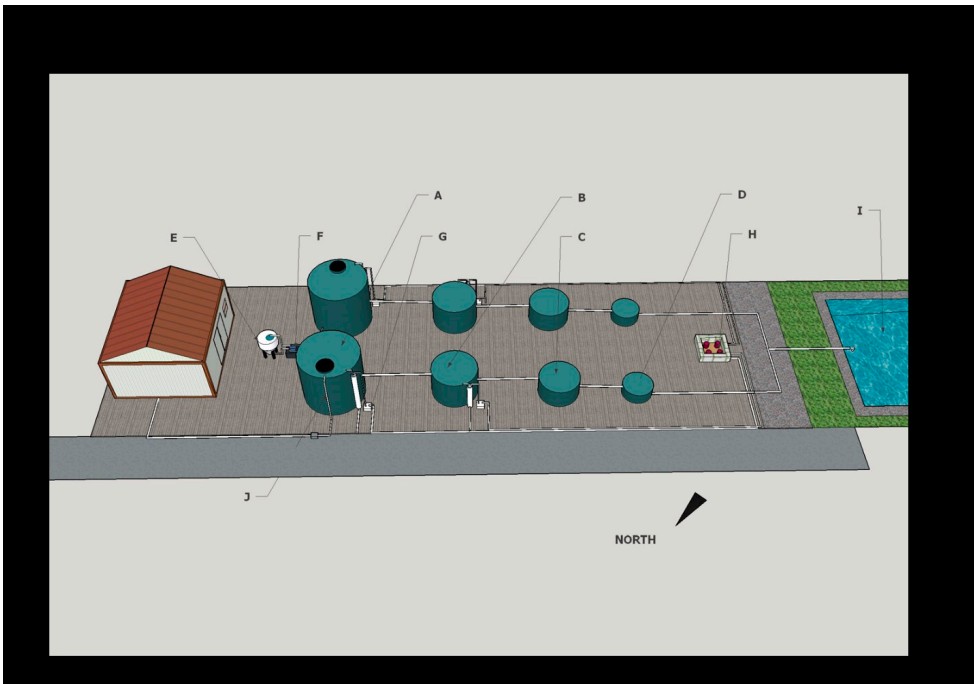

**Figure 1.** Anerobic digester systems layout. Lower row of tanks: sound-treated system; upper row of tanks: control system. (A.) Primary anaerobic digester, (B.) secondary anaerobic digester, (C.) partial nitrification tank, (D.) overflow tank, (E.) feed mixing tank, (F.) centrifugal trash pump, (G.) gas meter, (H.) gas flare pit, and (I.) water storage lagoon.

The holding tank outlets led to a polypropylene-lined lagoon designed for processing water recirculation back to the system and had dimensions of 14 by 7 m and an average depth of 4.75 m with a 30° slope, providing a capacity of approximately 125 m$^3$. During the period described, however, the 800-gallon partial aeration tank, which had been envisioned as a wastewater nutrient (NP) reduction tank, was not completed, and processed water was recirculated back to the system from the holding tank. Each tank was connected to the tank next in line by a 2-inch polyvinylchloride (PVC) pipe and electrically actuated, normally closed 2-inch full-port PVC ball valves (Valworx Inc., Cornelius, NC, USA) in-line with a mechanical float switch (Septic Products, Inc., Ashland, OH, USA). During operation, digestate levels were maintained via the ball valves/float switches at volumes of 9085-, 3785-, and 3028-L in the first three tanks, respectively. All pipe-to-tank connections were made via Uniseal® (US Plastic Corp., Lima, OH, USA) pipe-to-tank fittings.

One row of tanks was equipped with underwater loudspeakers and hydrophones, as described below, whereas no loudspeakers were placed in the control row of tanks, and hydrophones were installed in only the main control digester.

Each system was fed by an AMT model 316B-95 self-priming 120 V single-phase centrifugal pump (Gorman-Rupp, Mansfield, OH, USA). Wastewater was fed to the pump from an elevated cone-bottom HDPE feed tank with a capacity of 250 gallons (946 L). The wastewater was circulated back through the feed tank by means of a bypass circuit composed of a manual 2-inch PVC ball valve and pipe to ensure thorough mixing and afterwards fed to either wastewater system by means of another manual PVC ball valve and piping. An outlet for biogas venting was provided near the top of the primary and secondary digesters consisting of 0.5-in PVC pipe connected to an EKM model EKM-PGM-075 pulse output diaphragm gas meter, which recorded $1.0\text{-ft}^3$ (28.3 L) per pulse (EKM Metering, Santa Cruz, CA, USA).

The order in which the tanks were fed was alternated each week to account for residual feed in the supply lines to the primary digesters and ensure equal feeding to each row of digesters. The digesters were initially seeded on 1 May, 2018 with 750 L of swine waste from a facultative lagoon located on a swine feeding operation and from then on fed 22.7 kg of cracked corn in 750 L of wastewater from the 1150-L holding tank once weekly until July 17. The corn had a moisture content of 14.4% and volatile solids content of 97.1%.

From July 24 onwards, the digesters were fed 22.7 kg of cracked corn twice weekly until the conclusion of the experiment on October 18. Defatted soybean meal was substituted for cracked corn on June 21, as well as July 10 and 31. Due to low gas production, likely due to poor buffering and low pH, the digesters were fed 750 L of high-strength wastewater from a commercial digester located on a poultry farm in Central Kentucky on June 29. When fed once weekly, the primary digesters had a hydraulic retention (HRT) time of 12 weeks, and the secondary digesters had an HRT of five weeks.

*2.2. Audio Systems*

The audio speakers were Skar Audio FSX8 eight-in (20.3 cm) 4 Ω speakers rated at 175-W RMS (root mean square) power (Skar Audio, Tampa, FL, USA). The speakers were waterproofed by spray painting with Rustoleum® spray enamel (Rust-Oleum Corp., Vernon Hills, IL, USA) and then with GE Silicone I caulk (General Electric Co., Boston, MA, USA) diluted with petroleum-based lighter fluid to increase the fluidity of the coating and ensure complete coverage. In the primary sound-treated digester, two sets of speakers were placed. The first set was placed vertically approximately 30 cm above the bottom and from the sides of the tank to make the sound radiate transversely, whereas the second set was placed facing downwards near the middle of the tank 30 cm below the operating level of the digestate. Only one set of speakers were operated at a time.

Two Aquarian Model H1A hydrophones (Aquarian Audio & Scientific, Anacortes, WA, USA) were placed in the primary digesters of both the sound-treated and control wastewater treatment systems. One hydrophone was placed approximately 15 cm above the bottom center of the tank, while the other was placed in the center of the tank 1.0 m above the bottom of the tank. In the secondary sound-treated digester and partial-aeration tank, one hydrophone each was placed in the center of the tank.

Speaker and hydrophone lines were routed via a 2-in PVC conduit to a small building (4.8-m by 3-m) in which recording and amplifying equipment was kept. Audio amplifiers were Pyle Audio PTAU 55 stereo amplifiers rated at 120-W RMS per channel (Pyle Audio, Brooklyn, NY, USA). The hydrophones were connected by $\frac{1}{4}$" tip/sleeve connectors to a Behringer XR18 audio mixer (Behringer, Willich, Germany) interfaced to a computer operating under Windows® 7 Professional and running Reaper ver. 5.40/x64 rev 584a8e (Cockos Inc., New York, NY, USA). Live audio monitoring of the digesters was provided by 150 W Behringer Studio50usb-powered studio monitors. Sound was recorded with a sampling rate of 88,200 Hz with a 1200 ms buffer. These were converted to 320 kbps monaural MP3 files using Wavepad software ver. 6.63 (NCH software, www.nchsoftware.com). Hydrophone gain

was set to +20 dB on the mixer, with an input gain of 0 dB on each channel. Recording volume was set to 0 dB in Reaper.

Wave sound files were constructed using NCH Tone Generator ver. 3.26 (NCH Software) and Wavepad software. Constructed MP3 files consisted of single-frequency sine waves. To characterized audio phenomena in the digesters, sine waves of 100, 500, 1000, 2000, and 3000 Hz were used. The only sine wave used for extended periods in the digester was the 1000-Hz sine wave.

Musical files used for the experiments were stereophonic MP3 files obtained from Amazon (amazon.com) encoded at 256 kbps and converted into monaural format in Wavepad and amplified to boost loudness but below the point where clipping occurred (exceeding the dynamic range of the playback device). Files used were Movement 5 of Ludwig van Beethoven's Symphony No. 6 performed by The Royal Philharmonic Orchestra, the first movement of Nikolai Rimsky-Korsakov's Scheherazade Suite performed by The Chicago Symphony Orchestra, and The Rite of Spring composed by Igor Stravinsky and performed by The Philadelphia Orchestra. Sounds were played to the digesters at one-half volume on a 2-h-on, 1-h-off schedule controlled by a rotary timer.

Analysis of recordings made in the wastewater tanks were conducted using Wavepad and Audacity ver. 2.2.2 (https://www.audacityteam.org/). Power consumption by the amplifiers was measured with an Intertek power meter (Intertek Group PLC, London, UK).

### 2.3. Analyses

Weather data was obtained from the Kentucky Mesonet's Warren County weather station also located on the Western Kentucky University Farm (www.kymesonet.org). Biogas and wastewater quality analyses were performed weekly prior to feeding of the system, as described previously [18,19]. Samples for dissolved gas and biogas analysis were taken by means of a Luer fitting with 3-way connector and attached syringe. The samples were then injected into vials fitted with rubber septa without venting the samples. Water temperature, turbidity, conductivity, pH, and oxidation-reduction potential were measured using a YSI ProDSS water quality meter (YSI Inc., Yellow Springs, OH).

Statistical analysis was performed using SAS version 9.3 (SAS Institute, Cary, NC, USA). Mean comparisons of biogas yield and quality, as well as water chemistry parameters between the treatments, were performed using PROC ANOVA with Duncan's multiple range tests. Descriptive statistics were calculated using PROC MEANS.

## 3. Results

### 3.1. Acoustic Analysis of Digesters

Acoustic analyses of the digesters were performed during the summer of 2019 rather than the summer of 2018. This was because of a persistent 60-Hz hum that occurred in the recordings which we were unable to locate the source of until we disconnected the float switches controlling the PVC float valves. After identifying the source of the 60-cycle hum, power to the float switches and valves was shut off at the electrical breaker box except during digester feeding.

Initially, it was planned to compare the performance of speakers placed facing towards the bottom of the tank to those placed facing transversely at the bottom of the tank to determine if this would affect relative biogas production. We realized that this was not possible, since the system would exhibit hysteresis, and the amount of biogas produced would partially depend on the degree to which the previous sound treatment had affected sludge breakdown. Therefore, the pair of speakers that were operated were simply alternated week to week, and no significant difference in biogas production was noted due to speaker placement.

Previously [5], wastewater was exposed to 1000-Hz sine waves as a means of enhancing biogas production, as well as sludge breakdown. One of the interesting aspects of this study was that the bubble harmonics of the excitation frequency were evident in acoustic spectra of the wastewater, often followed by subharmonics symptomatic of incipient inertial cavitation. In the present study,

bubble harmonics at $2f_0$, $3f_0$, $4f_0$, ... , and, presumably, beyond the recording capability of our equipment, were also evident. We never observed the appearance of subharmonics, nor did we ever note the appearance of broadband noise indicative of cavitation cloud collapse.

In the previous study [5], biogas production was enhanced by 12 percent as compared to control digesters. We speculated that the effectiveness of sound treatment may have been limited by the speakers being placed in the sludge layer of the digester. The observation of widespread or chaotic cavitation may have been a direct consequence of the speaker placement, leading to dramatic acoustic effects but also limiting the effectiveness of the treatment.

Figure 2 illustrates the fundamental excitation frequency and first four harmonics for 100-, 500-, 1000-, 2000-, and 3000-Hz sine waves played at one-half volume, as observed in the primary anaerobic digester. These frequencies were chosen as being well within the optimum frequency range of the speakers (Supplemental Material). The 1000-Hz sine wave was the most powerful used, the fundamental frequency having a peak intensity of −1.5 dB relative to the full scale (dBFS) in our recordings, while the fundamental frequencies of 100, 500, 2000, and 3000 Hz had much lower intensities despite the speaker having a fundamentally flat frequency response from 500 to 4000 Hz in our tests. It is also interesting to note that the odd-numbered harmonics were more intense than even-numbered harmonics for all five sine waves.

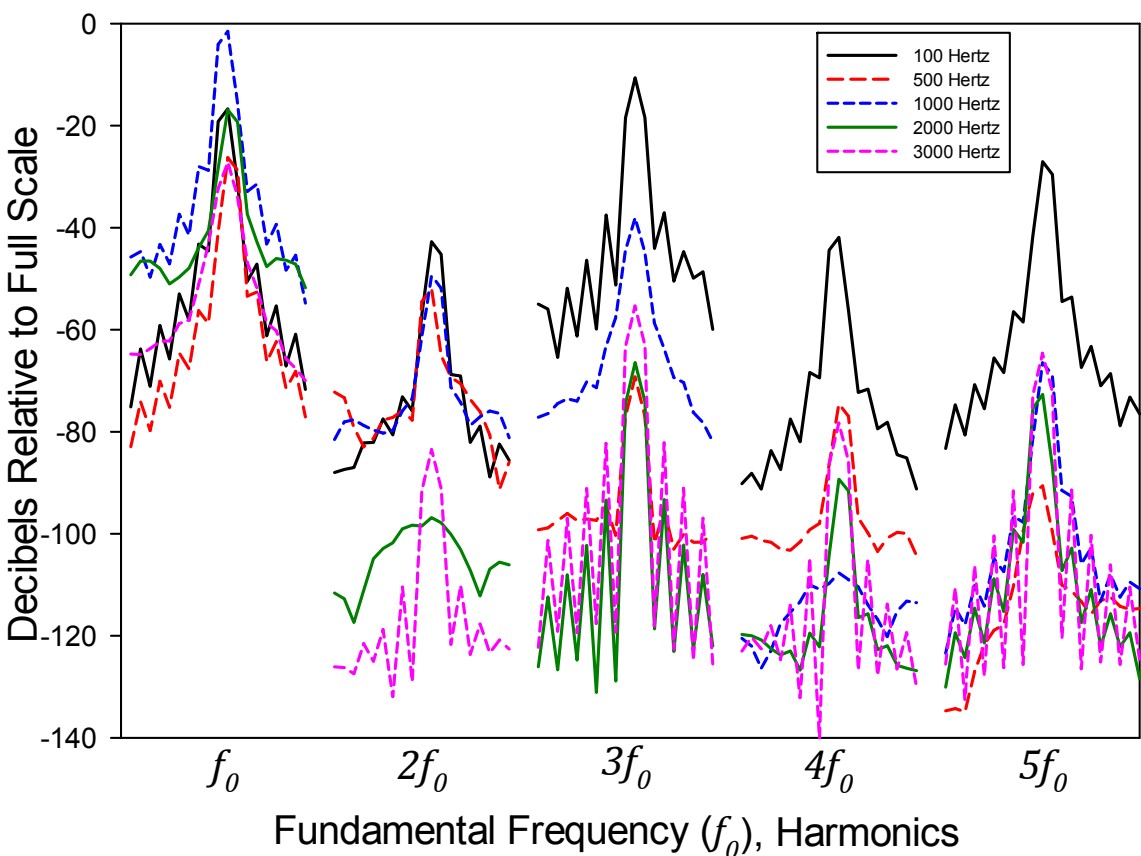

**Figure 2.** Fundamental excitation frequency ($f_0$) and first four bubble harmonics of sine waves as played to anaerobic digesters.

While the peak intensity of the fundamental frequency and harmonics were quite high in the digester, overall acoustic intensity at sonic frequencies was quite low and comparable to that of ambient sound in the digester due to the narrow frequency range of the fundamentals and harmonics. As stated in Loughrin et al. [5], we envision anerobic digestate as a gas-saturated liquid which, in addition to solvated gases, contains a continuum of bubble sizes, each of which will have a resonant frequency proportional to its radius. Therefore, in the expectation of affecting the vibrational state of the greatest

possible numbers of bubbles, we decided to use audio files with greater frequency coverage than was possible with single-frequency or even multiple-frequency sine waves. By varying the frequencies and intensities of the audio excitation, we also hoped to avoid the occurrence of standing waves, wherein the point of peak amplitude of a given frequency does not vary in space. This was particularly important considering the wavelength of the frequencies used. Assuming a sound velocity in salt water of 1400 m s$^{-1}$, a 1000-Hz sound wave would have a wavelength of 1.4 m. Since the primary anaerobic digesters only had a radius of approximately 1.2 m, rapid variations in wavelength would be more likely to transfer energy efficiently to the digester, since the presence of standing waves would be avoided and fixed points within the digester where sound amplitude were at maximum and minimum would be eliminated.

Music with broad frequency coverage was chosen for routine excitation of the digestate, rather than recordings of white or other broadband noise with similar or greater frequency coverage. When playing music to the digester, it was easier to spot the occurrence of cavitation events, since broadband noise obscured the frequencies induced by cavitation. We chose to use "classical" music, or what might be more properly termed orchestral compositions, since these works are often characterized by wide and often abrupt variations in amplitude and are harmonically complex. We felt that these characteristics would have a high likelihood of exciting a wide range of intrinsic bubble resonant frequencies and that variations in amplitude would allow us to more easily observe cavitation events. Musical compositions afforded higher amplitude in the range of 0–4000 Hz than did individual sine waves when played to the digestate, even when accounting for bubble harmonics of the excitation frequency (Figure 3). As a further benefit of using music rather than sine waves or broadband noise, we found that music was less likely to be transmitted through the concrete base of the system and be detected by the hydrophones in the primary control digester than were high-intensity, narrow-band sine waves. While some "cross-contamination" of sound did occur via transmission of sound from the sound-treated tank to the control tank, transmitted sounds recorded in the control digester were 30–40 dB less intense than in the sound-treated digester.

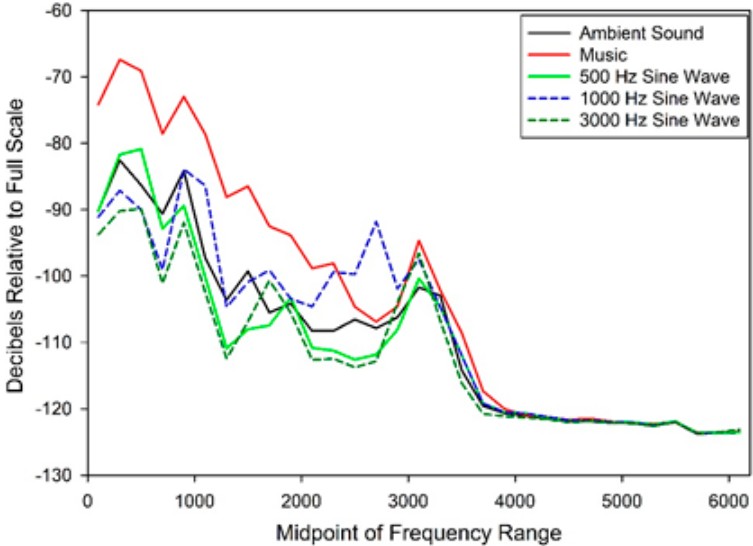

**Figure 3.** Audio spectra averaged every 200 Hertz for primary sound-treated anaerobic digesters exposed to various audio files. Peak audio intensities: ambient, −64.9 dB at 321 Hertz and Beethoven Symphony No. 6, Movement 5, −43.0 dB at 439 Hz. Sine waves: 500-Hertz sine wave, −15 dB; 1000-Hertz sine wave, −3.2 dB; and 3000-Hertz sine wave, −33.3 dB.

We did find that music, rather than sine waves or broadband white noise, was more likely to induce noticeable cavitational events. This seemed to confirm that the harmonic and amplitude variations of the music were more likely to perturb the bubbles than were broadband noise or sine waves. Some of

the cavitation events were quite violent, with amplitudes of as high as $10^4$ greater than the level of the music being played to the digester when the amplifier was at one-quarter volume. This represents a $10^4$ increase in energy input into the system, although without calibrated hydrophones, we were unable to quantify the intensity of these events.

Hydrophones were also installed in the primary control digester so that we might compare acoustics between the two tanks. After several weeks of exposure to sound on a two-h sound, one-h silence schedule, background recordings of the sound-treated primary digester were louder than those of the control primary digester (Figure 4). When sound was played to the primary sound-treated digester at half-volume, peak amplitude reached as high as −8 dBFS. When no sound was played to the primary digester, amplitude averaged roughly −58 dBFS but only about −87 dBFS in the control primary digester. There did not seem so much that there were more cavitation events in the sound-treated digester but that these events were louder than in the control digester. However, this may still be an indication of a greater number of cavitation events, since we do not know the distance at which the hydrophones were able to detect cavitation. Therefore, the apparent louder cavitation in the sound-treated digester may have been an indication of more cavitation inception or collapse occurring in the immediate vicinity of the hydrophone.

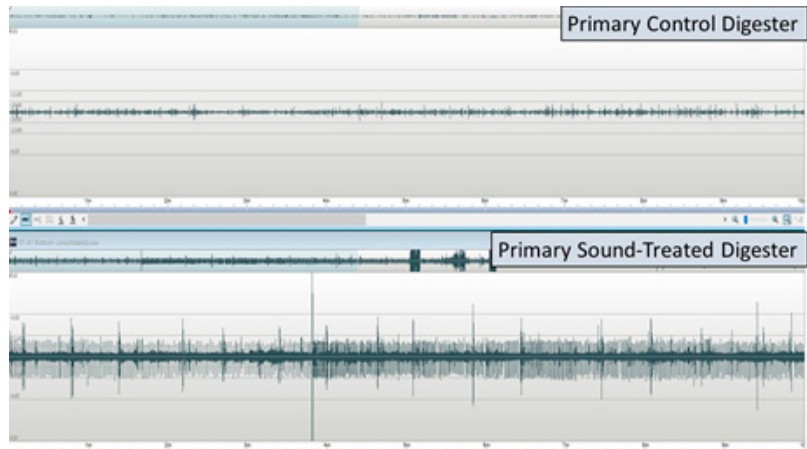

**Figure 4.** Background amplitude graphs of primary anaerobic digesters that had not been exposed to audio excitation (top) and which had been exposed to an ongoing treatment of two-h sound excitation, one-h silence.

In addition to the louder cavitation events in the sound-treated primary digester, the base amplitude was greater than that of the control digester, especially in the region from approximately 1500 to 3500 Hz. This may have been due to more bubble resonance in the sound-treated digester or simply due to a greater number of cavitation events that increase the background sound level of the sound-treated digester. It is interesting to speculate that a greater number, or greater average intensity, of cavitation event flows could be induced in the digester due to both cavitation inception and collapse and to drag-induced flows resulting from rising bubbles [20]. This would be similar to biogas recirculation, which has been used to provide some degree of mixing in anaerobic digesters [21].

The most likely explanation for the higher base amplitude of the sound-treated digester is greater metabolic activity than in the control digester. This could be due to a combination of physical and biological effects. Among the physical factors that could enhance biological activity in the digester are vibrational energy imparted to sludge, which would act to accelerate its breakdown; acoustic streaming, which would accelerate nutrient mixing; and accelerated sludge breakup due to acoustically-induced cavitation. All these factors could enhance microbial colonization and utilization of wastewater and sludge. Sound has been shown to enhance the growth of microorganisms such as *Brevibacillus parabrevis*, *Escherichia coli*, and *Saccharomyces cerevisiae* [22–24]. It is unclear whether accelerated microbial growth

in the presence of sound in these or our experiments is proximally due to physiological responses to the sound itself or greater nutrient availability due to mixing and matrix breakdown. Microbial community analyses are planned to determine if microbial numbers and/or populations were affected by the sound treatment.

Background sound levels in the secondary anaerobic sound-treated digester were also much lower than that of the primary sound-treated digester. Since gas production in the secondary sound-treated digester was only about 4% that of the primary sound-treated digester, this was not unexpected. Examples of recordings made from the digesters are given in the supplemental materials.

### 3.2. Digester Performance

Initially, the primary digesters were each seeded with approximately 750 L of swine waste obtained from the lagoon of a farrow to finish operation, along with 22.7-kg poultry litter and 4.5-kg cracked corn. From that point onwards, the digesters were fed 22.7 kg cracked corn or defatted soybean meal weekly until July 17, after which they were fed twice weekly. From April 10 through July 10, which we term the initial evaluation period, gas production from the digesters was low, averaging 3500 and 1910 L per week from the primary and secondary sound-treated digesters, respectively, and 2890 and 1250 L per week from the primary and secondary control digesters, respectively (Table 1). Gas quality was also poor, with $CH_4$ averaging only 39% and 59% of the combined total for $CH_4$ and $CO_2$ in the primary sound-treated and control digesters, respectively.

**Table 1.** Biogas production and quality and selected wastewater quality parameters during the initial evaluation period from April 10 through July 10.

| Parameter | Digester Stage-Treatment | | | |
|---|---|---|---|---|
| | Primary Sound | Secondary Sound | Primary Control | Secondary Control |
| Weekly Gas Production (L) [1] | 3520 ± 1310 a | 1910 ± 245 b | 2890 ± 681 a | 1250 ± 1300 b |
| | Biogas Concentration ($\mu$moles L$^{-1}$) [2] | | | |
| $CO_2$ | 12,700 ± 1840 a | 5450 ± 636 b | 12,000 ± 1950 a | 4450 ± 589 b |
| $CH_4$ | 8240 ± 3720 a | 18,600 ± 5120 a | 12,200 ± 4460 a | 13,500 ± 3920 a |
| | Digestate Concentration [2] | | | |
| pH | 6.30 ± 0.06 a | 6.98 ± 0.03 b | 6.33 ± 0.06 a | 7.34 ± 0.02 b |
| Chemical Oxygen Demand (mg L$^{-1}$) | 4700 ± 820 a | 1310 ± 356 b | 4370 ± 762 a | 1090 ± 227 b |
| $HCO_3^-$ (mM) | 2.53 ± 0.60 a | 4.11 ± 0.83 a | 3.26 ± 1.03 a | 3.95 ± 0.79 a |

[1] Data represent the mean of 71 determinations ± standard error of the mean. Within a row, means followed by the same letter are not significantly different by a Duncan's multiple range test at p = 0.05; [2] Data represent the mean of ten determinations ± standard error of the mean. Within a row, means followed by the same letter are not significantly different by a Duncan's multiple range test at p = 0.05.

This may have been due to inadequate microbial seeding of the digesters but was more likely the result of poor bicarbonate buffering and the resultant low pH. Bicarbonate buffering averaged only 2.53 and 3.26 mM in the primary sound-treated and control digesters, respectively, during this period, and pH averaged only 6.30 and 6.33. Gas quality was much higher in the secondary sound-treated and control digesters, with $CH_4$ averaging 77% in the secondary sound-treated digester and 75% in the secondary control digester. Bicarbonate buffering and pH were greater in the secondary digesters, which largely accounted for the better gas quality; on the other hand, the digesters were designed to retain as much sludge in the primary tanks as possible so that we could evaluate the effects of sound on sludge degradation. Due to this, gas production in the secondary sound-treated and control digesters averaged only 54% and 43% that of their respective primary digesters during this period.

On June 29, 750 L of digestate from a commercial thermophilic digester located on a poultry farm was added to each of the primary anaerobic digesters. This was done both for additional microbial seeding and to raise the pH of the digesters. This digestate had a high buffering capacity with $HCO_3^-$, averaging 341 ± 100 mM, and with an average pH of 7.32 ± 0.14.

After feeding of digestate from the thermophilic digester, gas production in the primary sound-treated digester increased by 895% and decreased by 58% in the secondary sound-treated digester, whereas gas production increased by 368% in the primary control digester and deceased by 25% in the secondary control digester (Table 2). The increase in gas production in the primary digesters was likely due both to microbial seeding and enhanced $HCO_3^-$ buffering. As stated, in the initial evaluation period, $HCO_3^-$-buffering averaged 2.5 and 3.1 mM in the primary sound-treated and control digesters, respectively, and 17.3 and 14.4 mM after addition of wastewater from the commercial digester. We ascribe the decrease in gas production by the secondary digesters after the initial evaluation period to more complete digestion of wastewater in the primary digesters.

**Table 2.** Biogas production, quality, and dissolved biogas concentrations during the principal evaluation period from July 3 through October 18.

| Parameter | Digester Stage-Treatment | | | |
|---|---|---|---|---|
| | Primary Sound | Secondary Sound | Primary Control | Secondary Treatment |
| Weekly Gas Production (L) [1] | 18,900 ± 2510 a | 773 ± 104 c | 9050 ± 1660 b | 896 ± 117 c |
| Biogas Concentration ($\mu$moles $L^{-1}$) [2] | | | | |
| $CO_2$ | 11,100 ± 525 a | 6570 ± 268 b | 10,700 ± 511 a | 6070 ± 254 b |
| $CH_4$ | 38,200 ± 1100 a | 30,000 ± 3150 b | 36,800 ± 1860 a | 20,600 ± 2360 c |
| Digestate Concentration (Millimolar) [2] | | | | |
| $HCO_3^-$ | 17.3 ± 1.4 a | 17.4 ± 1.5 a | 18.5 ± 1.5 a | 17.8 ± 1.5 a |
| $CO_2$ | 2.0 ± 0.1 a | 1.3 ± 0.1 a | 2.3 ± 0.2 a | 1.5 ± 0.1 a |
| $CH_4$ | 6.5 ± 2.3 a | 8.6 ± 3.4 a | 7.9 ± 2.8 a | 5.6 ± 2.3 a |

[1] Data represent the mean of 103 determinations ± standard error of the mean. Within a row, means followed by the same letter are not significantly different by a Duncan's multiple range test at p = 0.05; [2] Data represent the mean of 14 determinations ± standard error of the mean. Within a row, means followed by the same letter are not significantly different by a Duncan's multiple range test at p = 0.05.

Whether due to increased buffering or microbial seeding, from July 3 to October 16, gas production from the primary sound-treated digester was over twice that of the primary control digester (Figure 5). This difference was seen even though the primary control digester, due to a more southerly exposure, had significantly higher water temperatures (29.4 ± 0.3 °C) than did the primary sound-treated digester (28.2 ± 0.3 °C); t (15) and *p* = 0.0002.

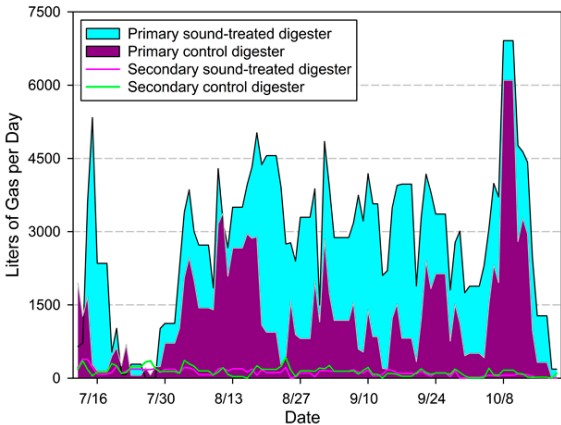

**Figure 5.** Daily gas production from anaerobic digester systems.

Water quality parameters in the primary sound-treated and control digesters were similar during both the preliminary and main evaluation periods. During the initial evaluation period of 12 weeks, total suspended solids (TSS) averaged 401 and 656 mg $L^{-1}$ in the primary sound-treated and control digesters, respectively, and 1000 and 1020 mg $L^{-1}$, respectively, during the main evaluation period (Table 3). During the preliminary evaluation period, the chemical oxygen demand (COD) averaged 4770 and 4370 mg $L^{-1}$ in the primary sound-treated and control digesters, respectively. COD concentrations decreased during the main evaluation period as biogas production increased. The higher COD concentrations in the sound-treated digesters as compared to the control digesters was likely due to enhanced sludge breakdown by sound, as noted in our previous research [5]. Obtaining representative sludge samples would have entailed opening the digesters and the mixing of the tanks for homogenization of the tank contents, so this analysis was not performed.

**Table 3.** Wastewater characteristics during principal evaluation period from July 3 through October 18.

| | Digester Stage-Treatment [1] | | | |
|---|---|---|---|---|
| Parameter | Primary Sound | Secondary Sound | Primary Control | Secondary Control |
| pH | 7.29 ± 0.03 b | 7.46 ± 0.02 a | 7.26 ± 0.02 b | 7.43 ± 0.02 a |
| Turbidity (FNU) [2] | 588 ± 89.1 a | 228 ± 38.1 c | 437 ± 54.9 ab | 290 ± 66.0 bc |
| Conductivity (µS $cm^{-1}$) | 8650 ± 372 a | 8290 ± 420 a | 8800 ± 352 a | 8210 ± 401 a |
| Oxidation Reduction Potential | −167 ± 4.7 a | −184 ± 4.5 b | −200 ± 3.9 c | −195 ± 9.2 c |
| | Digestate Concentration (mg $L^{-1}$) [3] | | | |
| Total Suspended Solids | 1000 ± 91.3 a | 320 ± 40.8 b | 1,020 ± 84.2 a | 370 ± 57.4 b |
| Chemical Oxygen Demand | 3200 ± 468 a | 1670 ± 219 bc | 2570 ± 321 ab | 1250 ± 180 c |
| $Na^+$ | 535 ± 34.2 a | 519 ± 31.9 a | 499 ± 34.6 a | 482 ± 25.6 a |
| $NH_4^+$ | 655 ± 61.2a | 584 ± 65.9 a | 665 ± 59.4 a | 622 ± 60.2 a |
| $Ca^{2+}$ | 39.4 ± 14.1 a | 39.4 ± 16.7 a | 32.1 ± 7.5 a | 34.4 ± 6.5 a |
| $Mg^{2+}$ | 35.1 ± 8.4 a | 39.6 ± 7.9 a | 32.1 ± 7.5 a | 34.4 ± 6.5 a |
| $PO_4^{3-}$ | 4.8 ± 2.9 a | 1.7 ± 1.1 a | 2.1 ± 1.2 a | 1.9 ± 1.1 a |
| $K^+$ | 641 ± 36.8 a | 595 ± 30.8 a | 618 ± 30.2 a | 561 ± 29.0 a |
| $Cl^-$ | 223 ± 12.9 a | 206 ± 17.0 a | 205 ± 14.8 a | 189 ± 15.7 a |

[1] Data represent the mean of 14 determinations ± standard error of the mean. Within a row, means followed by the same letter are not significantly different by a Duncan's multiple range test at p = 0.05; [2] Formazin Nephelometric Units; [3] Data represent the mean of 10 determinations ± standard error of the mean. Within a row, means followed by the same letter are not significantly different by a Duncan's multiple range test at p = 0.05.

The oxidation-reduction potential (ORP) was less negative in the sound-treated digesters than in the control digesters (Table 3). This was somewhat unexpected given that average COD concentrations were somewhat higher in the sound-treated digesters. The chemical oxygen demand test usually overestimates the bioavailability of solutes in wastewater, since it relies on strong oxidizing agents. Due to this, we believe that the ORP is a more reliable indication that the wastewater was more degraded in the sound-treated digesters than in the control digesters.

Gas quality during the principal evaluation period was good, with $CO_2$ averaging 11,000 $\mu$M $L^{-1}$ (489,000 $\mu$g $L^{-1}$) in the primary sound-treated digester and 10,700 $\mu$M $L^{-1}$ (471,000 $\mu$g $L^{-1}$) in the control primary digester. Methane during this same period averaged 38,200 $\mu$M $L^{-1}$ (613,000 $\mu$g $L^{-1}$) in the primary sound-treated digester and 36,800 $\mu$M $L^{-1}$ (591,000 $\mu$g $L^{-1}$) in the primary control digester. Again, this is similar to our previous research, in which biogas $CH_4$ concentrations were somewhat higher in sound-treated digesters [5]. Given greater pressure in the sound-treated digesters as a consequence of higher gas production, this was not unexpected.

Dissolved gas concentrations were similar during the preliminary and principal digester evaluation periods. During the principal evaluation period, aqueous $CH_4$ concentrations were 18% lower in the primary sound-treated digester than in the primary control digester. Although this difference was not statistically significant, it may have indicated some degassing of the digestate due to sound.

Dissolved gas concentrations in the secondary anaerobic digesters were likewise comparable to those of the primary control and sound-treated digesters. Most of the subsurface gas in the digesters is likely in the form of bubbles rather than in a dissolved state, however, so our method of estimating subsurface gases is likely to contain considerable measurement uncertainty by failing to obtain a representative sample of bubbles.

Gas production in the primary sound-treated digester averaged about 2700 L day$^{-1}$. This translates to a flux of about 40 $\mu$L cm$^{-2}$ min$^{-1}$ of biogas. With an active gas flux, it is hard to imagine that most of the subsurface gas does not exist in the form of bubbles with the flux largely released during bubble collapse at the digestate surface. These bubbles are subject to sonic manipulation, as we saw in our experiments.

### 3.3. Speaker Evaluation

A diagram of a loudspeaker is given as Figure 6. The speakers finished the trial with no noticeable loss in performance. At the end of the trial, however, it was noted that the dust cap/dome had become soft, and when the speakers were examined again several months after the trial, it was found that some of the speakers had failed due to short-circuiting from water infiltration. We have coated speakers that have continued to perform well for over six months in wastewater. These speakers, in contrast to the ones in this experiment, had had the port relief in the rear of the speaker filled completely with silicone and had especially heavy layers of silicone coating on the speaker spider and cone. In these speakers, no softness in the dust cap/dome was noted after prolonged use, so it is likely that filling the interior of the speaker with silicone is more likely to protect the voice coil and magnet from exposure to the wastewater. Filling the speakers completely with silicone does not appear to affect the performance of the speakers markedly; this is likely because the silicone elastomer used had enough elasticity to allow the speakers to transmit sound effectively. The only effect of the coating appeared to be some dampening of frequency response above 4000 Hz, which, in any case, was well above the optimum frequency response of the speakers.

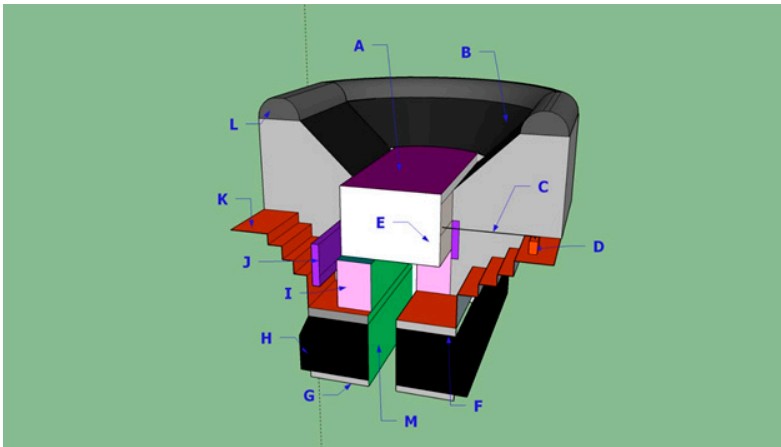

**Figure 6.** Schematic cutaway view of loudspeaker. (A.) Dust cap, (B.) speaker cone, (C.) terminal lead, (D.) terminal, (E.) coil former, (F.) magnet top plate/washer, (G.) magnet back plate, (H.) magnet, (I.) magnet pole piece, (J.) voice coil, (K.) speaker spider (suspension), (L.) speaker cone surround, and (M.) bass relief port.

We used GE Silicone I caulk for this experiment. When GE Silicone II caulk was used to coat the speakers, the speakers produced very little sound. This was likely due to a longer siloxane chain length and/or greater degree of cross-linking between the polysiloxane chains, reducing the flexibility of the coating. Thus, the silicone rubber used to coat the speakers needs to be chosen with care. We feel that waterproofing the interior of the loudspeaker and by using waterproofed materials for the diaphragm and spider, the speakers should perform well for extended periods in wastewater. A loudspeaker designed by the manufacturer to be waterproof and to operate efficiently underwater (i.e., transmit vibrational energy directly to the water rather than being enclosed in a housing) should be both relatively easy to implement and low-cost.

*3.4. Advantages of Acoustic Enhancement of Anaerobic Digestion*

Cracked corn was used to feed the digesters due to the time and expense of hauling animal wastes from distant producers. We felt this would serve as an adequate substitute for animal wastewater in our experiments and contribute to rapid-sludge formation in the digesters. The hydrolysis of sludge is usually the rate-limiting step in the anaerobic degradation of wastewater to produce biogas [25,26]. This has led to various pretreatments to disrupt sludge by mechanical means, such as ultrasonic disruption; treatment with surfactants to disrupt bacterial extracellular matrices [27]; alkaline chemical hydrolysis, which may or may not be coupled with ultrasonification [28,29]; and biological treatments such as incubation with bacteria that produce hydrolytic enzymes [30].

In the present study, sludge reduction without pretreatment in situ was attempted. This simplifies and reduces the expense of sludge handling. The results compare favorably to those of the ultrasonic pretreatment. For instance, Neis et al. [31] were able to improve biogas production by 30% over that of controls by using a 31-KHz ultrasonification pretreatment at a full-scale wastewater treatment plant, and Geng et al. [29] were able to improve biogas production by 69% by using pretreatment consisting of 20-kHz ultrasonification in conjunction with calcium oxide hydrolysis.

Cavitational collapse of bubbles with radii that are resonant at ultrasonic frequencies differs from that of bubbles resonant at sonic frequencies. Cavitational collapse induced at ultrasonic frequencies releases more energy than that induced at sonic frequencies, and temperatures as high as 6500 $^0$K have been measured [9]. These temperatures are not generated during cavitational collapse with large bubble radii. This should result in less deleterious effects on the microflora of the digester and, given the longer wavelength of the sound excitation, less sound attenuation [32]. Thus, while ultrasonification

would be the preferred method for treatment of sewage influent, sonification would be preferred in the larger volumes of a digester.

Another common practice in anaerobic digestion is to operate the digesters at either mesophilic (30–40 °C) or thermophilic (50–60 °C) temperatures to accelerate biogas production and reduce sludge volumes [33]. It has been estimated that the heating requirements for thermophilic digestion are approximately twice that of mesophilic digestion, though considerable savings may have been realized due to shorter wastewater retention times and smaller digester volumes [3]. In this study, biogas production was more than doubled over that of the control at an average wastewater temperature of 28.2 °C and a maximum recorded temperature of 32.1 °C. The energy required to raise the temperature of one of the primary digesters from 28.1 °C to the midrange of mesophilic operating conditions (35 °C), assuming 100 percent efficiency, would be approximately 62,700 kcal, with considerable energy required to maintain the elevated temperature.

In contrast, the power requirements for acoustic treatments are low. In our experiments, only about $72 \pm 46$, $62 \pm 30$, $68 \pm 0.3$, and $140 \pm 0.8$ W were consumed by the amplifiers at half-volume when playing The Rite of Spring; Beethoven's Symphony No. 6, Movement 5; white noise; and a 1,000-Hz sine wave, respectively. Average energy consumption of the stereo system operated at one-half volume was 68.4 W. The sound system was typically operated 16 h day$^{-1}$, which translates to approximately 1150 kcal day$^{-1}$.

The water temperature during the principal evaluation period did not go below 15.5 °C and that was just as the experiment was terminated, so it is unknown to what extent sound treatment of anaerobic digestate might act as a substitute for wastewater heating. Nevertheless, the results of this experiment clearly show that audio treatment of digestate has the potential to decrease heating requirements for anaerobic digesters.

## 4. Conclusions

Acoustic treatments of digestate offer an economical means of enhancing biogas production. The increase in gas production is likely due to several factors that include acoustically-induced cavitation and vibrational energy acting together to break down wastewater solids. In addition, the increased release of bubbles from sludge may help to enhance mixing.

A further benefit is a reduction in sludge volume, reducing the need for pretreatment of feedstocks. Furthermore, in situ acoustic treatments could be implemented in existing biogas plants with little modifications. Research is needed, however, to see if sonic treatments of anaerobic digesters can be effective at commercial scales.

**Supplementary Materials:** The following are available online at http://www.mdpi.com/2076-3298/7/2/11/s1: Figure S1: Frequency response of coated and uncoated speakers. Notes on MP3 recordings.docx: explanatory notes for MP3 recordings, MP3 file 1:3 min of primary control digester amplified twice at 400% 3 d after feeding, MP3 file 2:3 min of primary sound-treated digester amplified twice at 400% 3 d after feeding, MP3 file 3: August 1 primary control digester amplified once at 200%, MP3 file 4: August 1 primary sound-treated digester amplified once at 200%, MP3 file 5: background of primary control digester midday—day after feeding bottom hydrophone, and MP3 file 6: background of primary sound-treated digester midday—day after feeding bottom hydrophone.

**Author Contributions:** Conceptualization: J.L.; methodology, J.L., S.A., and N.L.; investigation: J.L., S.A., and N.L.; writing—original draft preparation: J.L.; writing—review and editing: J.L., S.A., N.L., and K.S.; supervision: J.L. and S.A.; and funding acquisition: J.L. and K.S. All authors have read and agree to the published version of the manuscript.

**Funding:** Funding was provided by the Agricultural Research Service (Grant No. 5040-12630-006-00D).

**Acknowledgments:** The authors thank Michael Bryant and Zachary Berry (USDA-ARS) for technical assistance and Sean Thomas (Musician's Pro, Bowling Green, KY, USA) for setup of recording software. The use of trade, firm, or corporation names in this website is for the information and convenience of the reader. Such use does not constitute an official endorsement or approval by the United States Department of Agriculture or the Agricultural Research Service of any product or service to the exclusion of others that may be suitable.

**Conflicts of Interest:** The authors declare no conflicts of interest. The funders had no role in the design of the study; in the collection, analyses, or interpretation of data; in the writing of the manuscript; or in the decision to publish the results.

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
