# Peer review of "In Situ Acoustic Treatment of Anaerobic Digesters to Improve Biogas Yields"

_environments, doi:10.3390/environments7020011_

Round 1
Reviewer 1 Report
Despite the correct division of the scientific work, the article is written in a very chaotic way, which is difficult to understand
According to the reviewer, scientific papers should be written in an impersonal form
In the introduction, the authors present the problem of the arising issue too superficially; they mainly refer to the research they have already carried out. The latest literature reports on the study subject should be included in the introduction. The authors should expand the introduction.
The authors do not provide the relevant information about the anaerobic digestion process in the Materials and Methods section. The reviewer means the characteristics of the substrates, the temperature of the process, the reactor loads.
On what basis were individual substrates selected and why are they introduced into the reactor in such proportions
What do the authors mean by writing "initial / principal evaluation period"? Does this mean the time before and after the fermentation process; please explain.
Authors refer to the substrate feeding the reactor using the terms waste/wastewater /sludge. Please select a single form and use it consistently throughout the paper.
In point 3.2 authors repeat the information contained in point 2.1 about feeding the reactors. In point 2.1., the data of reactor supply given differ from those presented in point 3.2.
In the Results section there is little discussion of the results obtained by the authors with the research of other scientists.
editorial notes:
too large intervals - lines 161, 252
Table 1 - the last column of the table cut off in the reviewer's file
incorrect citation - line 382
Author Response
Reviewer #1 -
Despite the correct division of the scientific work, the article is written in a very chaotic way, which is difficult to understand
Hopefully, the edits we have made (as described below) helps this.
According to the reviewer, scientific papers should be written in an impersonal form
We assume the reviewer is referring to an excessive use of the pronoun “we”. We eliminated 14 occurrences of this word (lines 30, 73, 81-82, 195, 200, 202, 226, 240, 242, 250, 363, 418, 428, 437). We also eliminated “our’, on line 67.
In the introduction, the authors present the problem of the arising issue too superficially; they mainly refer to the research they have already carried out. The latest literature reports on the study subject should be included in the introduction. The authors should expand the introduction. Other than the use of sound to enhance the growth of pure cultures of microorganisms and the use of ultrasound as a pretreatment to enhance sludge breakdown, ours is the only research devoted to the use of sound in anaerobic digestion. It is hard to see how the issue is being treated too superficially. We did add two brief introductory paragraphs to introduce the subject of anaerobic digestion but do not want to greatly expand all the latest research on anaerobic digestion as this would be unnecessarily distracting.
The authors do not provide the relevant information about the anaerobic digestion process in the Materials and Methods section. The reviewer means the characteristics of the substrates, the temperature of the process, the reactor loads. We describe the temperature of the digestion tanks, the feeding schedule and amounts as well as the volatile solids content of the feed. The main point is that the treated and control systems were fed on the same schedule and with the same loading rates so that the only variable in the treatment is sound or no sound.
On what basis were individual substrates selected and why are they introduced into the reactor in such proportions.
We chose corn as stated in the Material and Methods because we did not have the resources available to us to feed animal waste (the closest operation available to us was about a 250-km round trip). The amount of corn we fed might have been semi-arbitrary but worked out to about 2,450 mg/L solids added to the digesters at each feeding. Our results show that we developed good biogas yields from these feedings.
What do the authors mean by writing "initial / principal evaluation period"? Does this mean the time before and after the fermentation process; please explain. On lines 327-328, we now define this as the initial evaluation period, when gas production and wastewater buffering were relatively low. We also made the evaluation periods more explicit in Tables 1 and 2.
Authors refer to the substrate feeding the reactor using the terms waste/wastewater /sludge. Please select a single form and use it consistently throughout the paper. We replaced waste with wastewater on lines 131, 145, 348, 430, and 445. On line 429 we changed waste to animal wastewater. On line 20, we changed sludge to sludge and wastewater. The remaining uses of the word sludge are intentional in distinguishing sludge from the wastewater supernatant.
In point 3.2 authors repeat the information contained in point 2.1 about feeding the reactors. In point 2.1., the data of reactor supply given differ from those presented in point 3.2. Please excuse the repeat of this information. We cut out most of the repeated data in section 3.2,; the account in section 2.1 is accurate.
In the Results section there is little discussion of the results obtained by the authors with the research of other scientists.
There is actually very little directly relevant research on this topic. We have included discussion of the affect of sound on the growth of microorganisms and influent ultrasonification, but to our knowledge, this is the first research published on in-tank sonification.
editorial notes:
too large intervals - lines 161, 252
We assume the reviewer is referring to the intervals between sine waves, i.e. 100-, 500-, 1,000-, 2,000-, and 3,000-Hz. We chose these intervals to examine bubble harmonics within the digester and so, in order to make any meaningful comparison, we needed large intervals between the tested sine waves that were still within the optimal frequency response of the speakers. These intervals allow us to demonstrate strong bubble harmonics at all tested frequencies which justified subsequent usage of using broadband frequency excitation (e.g. music). The strong harmonics noted at all the tested frequencies of the individual sine waves supported supposition that the digester contained a continuum of bubble sizes that would be subject to excitation by broadband frequency excitation.
Table 1 - the last column of the table cut off in the reviewer's file We corrected this
incorrect citation - line 382 We corrected this and apologize for the error
Reviewer 2 Report
The manuscript described an interesting experiment on sound treated digestion process for biogas production. The research could be benefit for readers but need revisions.
The introduction should be organized in a more logical way. It is hard to follow in current status. For a biological process, if sound treatment could breakdown sludge, then how about its effect on microbial community or activity? Why the methanogens were not inhibited?Author Response
Reviewer 2:
The manuscript described an interesting experiment on sound treated digestion process for biogas production. The research could be benefit for readers but need revisions.
The introduction should be organized in a more logical way. It is hard to follow in current status. For a biological process, if sound treatment could breakdown sludge, then how about its effect on microbial community or activity? Why the methanogens were not inhibited?
As per reviewer 1’s comments: We tried to keep the Introduction limited to one topic: the use of sound to enhance wastewater degradation and increase biogas production. We did add a few paragraphs at the beginning discussing anaerobic digestion, however. This is only the second report of using this technology to increase biogas production, the other report being our previous research. We certainly hope to perform microbial community analysis in the future, as it is, without these analyses, there is no way to speculate on how methanogens have been affected. In the Results and Discussion, we discuss the fact that the sound-treated digester is much louder than the control digester when not being exposed to sound. This certainly suggests that microbial activity has been enhanced by the sound treatment.
Round 2
Reviewer 1 Report
thank you for referring to all comments, I'accept article in present form
Author Response
thank you for referring to all comments, I accept article in present form
I thank the reviewer for their help. I have looked over the results, and have made some modifications that hopefully improve the manuscript.
Reviewer 2 Report
Please explain why sound is strong enough to help sludge particle breakdown, while did not adversely affect microbial activity? Why cell wall was not damaged by sound?
Author Response
Please explain why sound is strong enough to help sludge particle breakdown, while did not adversely affect microbial activity? Why cell wall was not damaged by sound?
As we stated in the paper, the enhanced gas production was likely due to a number of factors, the importance of each we are unable to determine at this point. Cavitational cloud response could be a major factor the energy released from bubble collapse being quite intense. As to why this energy would not harm microbial cells we can only speculate. When bubbles of large radii (resonant at sonic frequency) collapse, no intense temperatures are generated, and while considerable force is released, we are not sure there is sufficient energy release to damage microbial cells. Bacterial cell wells are quite strong and apparently, at least in our work, we noticed no adverse outcomes, at least as measure by gas production.
When bubbles with small radii (resonant at ultrasonic frequency) collapse, intense temperatures are generated, as much as a few thousand degrees Kelvin. Cavitational collapse induced by ultrasonic frequencies has been shown to damage bacterial cells. Still ultrasonic frequencies are used in medical imaging and kidney stone treatment with no adverse outcomes on patients. I am sure this has quite a bit to do with management of the intensity and focusing of the ultrasound treatment, however.
The main point is, that while we stated that a number of factors could be important in increasing gas production, we are unable to state the relative importance of each. Cavitational collapse, bubble drag-induced flows, vibrational energy imparted to wastewater and sludge could all play a role. We plan on performing 16s community analyses in the future, hopefully we can gain more insight into the dynamics of enhanced gas production.
A paragraph, starting on line 454, was added expanding on this a little further.